# Effect of Different Desensitizers on Shear Bond Strength of Self-Adhesive Resin Cements to Dentin

**DOI:** 10.3390/bioengineering9080372

**Published:** 2022-08-07

**Authors:** Alejandro Elizalde-Hernández, Louis Hardan, Rim Bourgi, Cristina Pereira Isolan, Andressa Goicochea Moreira, J. Eliezer Zamarripa-Calderón, Evandro Piva, Carlos Enrique Cuevas-Suárez, Walter Devoto, Ahmed Saad, Patrycja Proc, Monika Lukomska-Szymanska

**Affiliations:** 1Graduate Program in Dentistry, Federal University of Pelotas, Pelotas 96015-560, RS, Brazil; 2Department of Restorative Dentistry, School of Dentistry, Saint-Joseph University, Beirut 1107 2180, Lebanon; 3School of Dentistry, Federal University of Jequitinhonha and Mucuri Valleys (UFVJM), Diamantina 39803-371, MG, Brazil; 4Academic Area of Dentistry, Autonomous University of Hidalgo Sate, Pachuca 42083, Mexico; 5Dental Materials Laboratory, Academic Area of Dentistry, Autonomous University of Hidalgo State, Circuito ex Hacienda la Concepción S/N, San Agustín Tlaxiaca 42160, Mexico; 6Independent Researcher, 16030 Sestri Levante, Italy; 7Department of Restorative Dentistry, College of Dentistry, Al-Bayan University, Baghdad 100013, Iraq; 8Department of Pediatric Dentistry, Medical University of Lodz, 92213 Lodz, Poland; 9Department of Restorative Dentistry, Medical University of Lodz, 92213 Lodz, Poland

**Keywords:** bond strength, contact angle, dentin, desensitizers, resin cements

## Abstract

The sealing and bonding characteristics of luting cements may be affected by the application of desensitizers containing ingredients that induce chemical interaction with dentin organic matrix. This study evaluated the effect of different desensitizers on the immediate and long-term shear bond strength (SBS) of a self-adhesive resin cement (SARC) to dentin. Healthy bovine dentin specimens were used for the study. Gluma^®^ Desensitizer, Desensibilize Nano P^®^, and Soothe^®^ desensitizer were used in study groups, while the control group did not receive any treatment. Next, SARC (RelyXTM U200) in cylindrical mold was applied to the sample surface. All specimens were stored at 37 °C for 24 h or six months and tested for SBS. Additionally, water contact angle was measured using an optical tensiometer. Results were analyzed by analysis of variance and Student-*t* tests (*p* ˂ 0.05). Application of the different types of desensitizers had no significant influence on immediate or long-term SBS of SARC to dentin (*p* > 0.05). Differences for water contact angle were not statistically significant among the tested groups (*p* = 0.450). Within the limitations of the present study, it can be concluded that the application of the different types of desensitizers had no significant influence on the SBS of a SARC to dentin.

## 1. Introduction

Dentin hypersensitivity following tooth preparation for crown reduction is a common problem in dentistry, however very rarely documented [1]. Previous in vitro study revealed that reducing the amount of water cooling or increasing the air pressure and load during cavity preparation increases the temperature of the pulp chamber, which could result in hypersensitivity or even pulp necrosis [2]. Other factors, such as aggressive tooth grinding or preparation, preparation time, preparation thickness, method of manufacture and adjustment of provisional, bacterial contamination, and dehydration of dentin, are also implicated [3].

Brannström’s hydrodynamic theory posits that dentinal hypersensitivity is promoted by external stimuli, such as thermal, tactile, chemical, or osmotic pressure, that cause movement of intratubular dentinal fluids in exposed dentin [4]. Such fluid movement is capable of exciting the nerve fibers that induce hypersensitivity or pain [5].

The treatment of dentin sensitivity after tooth preparation is intended to offer immediate and permanent pain relief. However, many of these treatments may be unsatisfactory, since most of the conventionally used desensitizers are related to the occlusion of the dentinal tubules without considering the causal factors that have triggered the problem [6]. Therefore, the initial therapeutic strategies should aim to eliminate predisposing factors, such as abrasion, erosive components, and abfraction, thus avoiding recurrence of symptoms [7]. The therapeutic procedures options for reducing dental hypersensitivity are based on substances that depress transmission, such as potassium salts or potassium nitrate, substances that occlude the dentin tubules by stimulating mineral deposits, such as fluorides, oxalates, varnishes, adhesive resins, Bioglass^®^, and Portland cement, while low power (Helium-Neonium; He-Ne, Aluminum Gallium Arsenide; AsGaAl) and high power laser treatments (Neodymium Yttrium Aluminum Granate; Nd:YAG, carbon dioxide;CO_2_) are considered therapeutic treatments [8]. 

Clinical studies have evaluated the use of fluorides for the treatment of tooth sensitivity. Kielbassa et al. [9] evaluated two types of a commercial fluoride lacquers, one that contained 6% sodium fluoride (NaF) and calcium fluoride (CaF_2_), and another one used as a control that only consisted of 6% NaF. In their subsequent evaluations at six and 12 months, the hypersensitivity scores decreased after treatment. There was a pain relief, however none of the treatments completely eliminated the problem. Easily soluble NaF provides a rapid release of fluoride ions, which are converted to CaF_2_ on the tooth surface to effectively assist in remineralization. The CaF_2_ was slowly soluble in saliva, which would justify the transient action of a chemical barrier that provides a lasting retention on the surface of the tooth. This guarantees a long-lasting fluoridation. It was concluded that a lacquer containing CaF_2_/NaF in treating dentin hypersensitivity is effective in the initial reduction of dentin hypersensitivity. The combination of CaF_2_/NaF can be recommended for clinical use. On the other hand, when comparing the desensitizing effects of a gallium–aluminum–arsenide (GaAlAs) laser and NaF, an immediate reduction in the visual analog scale (VAS) score was observed. However, the NaF group showed an increase in the VAS scale at three and six months in comparison to one week and one month. The authors concluded that GaAlAs laser irradiation was effective in treating tooth sensitivity and was considered a more comfortable and faster procedure when compared to traditional treatment [10]. Nowadays, these are the main mechanisms through which desensitizers adequately manage pain. 

The clinical success of an indirect restorative procedure depends on several factors; however, cementation technique is a crucial step for long-term clinical success, which favors retention and prevents micro-leakage, secondary caries, and restoration loss [11]. Cementation could be performed using either conventional water-based cements or resin-based cements; among the latter, self-adhesive resin-based cements have been introduced into the market to facilitate the cementation of fixed restorations. These cements do not require any pretreatment of the tooth surface, thus reducing the application time and the technique sensitivity [12]. The self-adhesive resin cements (SARCs) are able to effectively diffuse and decalcify the underlying dentin due to increasing viscosity occurring after paste-to-paste mixing (owing to an acid-based reaction) [13]. Additionally, a greater contact with dental tissues to react with hydroxyapatite is observed, possibly resulting in an enhanced monomer dentinal interaction with the dental tissues [13]. Moreover, a high hydrophilicity enhances wetting of the tooth surface and a low pH—etching of the tooth substrates [14]. As a consequence, dentin and enamel demineralization takes place. Next, the carboxylic and phosphoric acid-groups of the modified methacrylate monomer present in the SARCs interact with the calcium from the hydroxyapatite (enamel and dentin) [12,15]. As the adhesion to the tooth structure has been established the acidity of the SARCs is being neutralized (from 2.8 to 7.0 after 24 h) [13]. These materials are structurally similar to compomers, the main difference being the concentration of acid monomer. Most of these SARCs may contain somewhat lower filler particles compared to compomers. Therefore, hydrated substrates more efficiently facilitate the ionization of acid monomers followed by acid-base neutralization reactions involving the tooth and the basic filling. For this reason, SARCs may demonstrate adhesion to dentin. In summary, due to the composition of SARCs that do not contain water, the dentin surface treatment must not dry out excessively before the application of this cement. However, over-wetting of the adherent dentin surface can hinder polymerization and reduce the integrity of the bonding interface. For this reason, the chemical interactions between the functional acid monomers of the SARCs and the dental substrate (dentin and enamel) are important mechanisms for adhesion. In addition, the ability of self-adhesive systems to release fluoride has been investigated, and they were found not to provide postoperative sensitivity [16].

The sealing and bonding characteristics of these luting cements may be affected by the application of desensitizers containing ingredients that induce chemical interaction with dentin organic matrix. Nevertheless, few studies have evaluated the effect of desensitizers on the shear bond strength (SBS) of the self-adhesive resin cement (SARC) to dentin. Therefore, this study aimed to evaluate the effect of three different desensitizers on the immediate and long-term SBS of SARC to dentin. The null hypothesis tested was that the application of desensitizers will not affect the immediate or long-term SBS of a SARC to dentin.

## 2. Materials and Methods

### 2.1. Experimental Design

In this work, the bond strength between a SARC (Rely X U200; 3M ESPE, St. Louis, MI, USA) and bovine dentin was evaluated according to the following factors: (1) previous application of a desensitizer agent at three levels (Gluma^®^ Desensitizer, Nano P^®^, and Soothe^®^); and (2) storing time at two levels (24 h and 6 months). These aging times were chosen following the directions of the ISO/TS 11406 International Standard [17]. A group without the application of a desensitizer agent was used as control. The chemical composition and application protocols of the desensitizing agents used in this study are described in Table 1. 

### 2.2. Specimen Preparation

Healthy bovine incisors were sectioned at the cemento-enamel junction using a low-speed motor (MotorTurbo & E-ASP1, Eighteeth, Changzhou, China) with cooling. Crowns were embedded in cylindrical plastic molds using cold-cure acrylic resin, which allowed the buccal enamel surface to be exposed. Next, the buccal enamel was abraded with an orthodontic grinder until a flat medium dentin surface was exposed. Then, samples were standardized by polishing with 600 grit silicon carbide sandpaper for 1 min. Afterward, each sample was examined under a light stereomicroscope at a magnification of 40× to verify the exposed dentin. 

The desensitizers were applied four times with one-week intervals on the surface of samples according to manufacturer’s instructions. Between each surface treatment, specimens were stored (for 7 days) in artificial saliva at the temperature of 37 °C, thereby simulating clinical conditions. The control group did not receive any treatment was stored in artificial saliva at 37 °C for four weeks.

Next, specimens were washed with distilled water and excess dentin moisture was removed. Elastomer molds with two cylindrical opening (1.5 mm diameter, 0.5 mm thickness) were placed at the center of the sample. The SARC Rely X U200 was applied (according to manufacturer instructions) and polymerized for 40 s with Curing Pen (Eighteeth, Changzhou, China) at 1000 mW/cm^2^.

The sample size (*n* = 10) was estimated based on the data of other study [18], considering a comparative study design of four independent groups, a minimum detectable difference in SBS of 3.1, a standard deviation (SD) of 1.9, a power of 0.8, and α = 0.05. 

### 2.3. Shear Bond Strength Test

Samples in each study group were tested after aging in distilled water at 37 °C for 24 h and for 6 months. The plastic molds containing the specimens were fixed to a microshear test device (Odeme Dental Research, Luzerna-SC, Brazil). Next, a thin steel wire (0.2 mm diameter) was looped around the cylinder and aligned with the bonding interface. The steel wire was then pushed upwards applying a tensile force (Figure 1). SBS test was conducted at a crosshead speed of 0.5 mm/min until failure in a universal testing machine (EMIC^®^, DL 500; São José dos Pinhais, Brazil). SBS (in MPa) was calculated by dividing the maximum force achieved by the area (1.77 mm^2^) of the bonded specimen. 

Additionally, the bonding surface was analyzed with a stereomicroscope (50×) to determine the failure mode: adhesive, cohesive in dentin, cohesive in resin, and mixed [19].

### 2.4. Contact Angle

The water contact angle of the dentin after application of the desensitizing agents was measured with an optical tensiometer (Theta Lite TL101, Biolin Scientific Inc; Stockholms Lan, Finland) following a sessile drop method. For each study group, the dentin surface was prepared as previously described. Standardized drops of the distilled water (5 μL) were directly dispensed onto the dentin surface. Immediately after placing the drop onto the dentin surface, a dynamic reading of the right and left contact angle was measured in real time with One Attension software (Biolin Scientific Inc, Stockholms Lan, Sweden) using 20 frames per second for 20 s. The contact angle (°) was estimated as the mean between the right and left readings (*n* = 3).

### 2.5. Statistical Analysis

Data were analyzed using SigmaPlot 14.0 software at a statistical significance of α = 0.05. Approximate normality of data distribution was determined by Kolmogorov–Smirnov and Shapiro–Wilk tests. For each storage period, one-way analysis of variance, followed by Tukey post-hoc test was used to detect significant differences between the desensitizer groups. The effect of aging condition was analyzed by two-sample Student’s *t*-test.

## 3. Results

### 3.1. Shear Bond Strength and Failure Mode

Means and standard deviations of SBS values at 24 h and six months aging are shown in Figure 2. At 24 h, the only statistically significant difference detected was for the comparison between Gluma^®^ Desensitizer and Soothe^®^ SDI (*p* = 0.049). When compared to the control, none of the desensitizers showed statistically significant differences (*p* > 0.05). The highest value was observed for Gluma^®^ Desensitizer (7.8 ± 0.99), while the lowest was for Soothe^®^ SDI (5.95 ± 1.26). At six months, the lowest SBS value was obtained for the Nano P ^®^ desensitizer group (3.23 ± 0.73 MPa) and the highest value was observed for Gluma^®^ Desensitizer (4.55 ± 0.62). However, the only statistically significant difference in SBS at six months was found between Gluma^®^ and Nano P^®^ (*p* = 0.033). The comparisons of the SBS values after 24 h and six months of aging by Student’s *t*-test showed statistically significant differences in the SBS means between all the groups (*p* < 0.05) (Figure 3). The failure mode in all study groups was adhesive (Figure 4).

### 3.2. Contact Angle

Contact angle measurements are shown in Figure 5. Differences in water contact angle were not statistically significant among the study groups (*p* = 0.450).

## 4. Discussion

In this study, the immediate and long-term SBS of a SARC to bovine dentin after application of different desensitizers was evaluated. The results suggest that the use of the desensitizers had no influence on the SBS. Therefore, the hypothesis that the application of desensitizers will not affect the immediate or long-term SBS of a SARC was not rejected.

It has been previously reported that the SBS of resin cements depends on several factors, such as the type of dentin (coronal, apical, caries-affected, and sclerotic), preparation depth (superficial or close to the pulp), and tooth surface management [20,21,22,23]. For SARCs, no conditioning of dentin with a bonding agent is needed. Therefore, the surface properties of the tooth substrates play an important role in adequate adhesion. Actually, the adhesion of SARC to dentin and several restorative materials are acceptable and comparable to other multi-step resin cements, but the adhesion to enamel seems to be a weak link in regard to their bonding properties. These cements are characterized by the presence of hydrophilic acidic monomers that bind directly to the wet dentin surface [16]. Their ability to adhere to dental structures mainly depends on the monomer infiltration into the tissues and the formation of a resin-infiltrated layer [16], which also depends on the presence of hydroxyapatite to adhere to the dental structures [12,14].

Considering that desensitizers can alter the characteristics of the dentin surface, it could be hypothesized that they have also the potential to alter the SBS of SARCs. A limited number of studies have evaluated the impact of the desensitizers on self-adhesive resin-dentin interface, reporting contradictory results [24,25,26,27,28]. For example, Stawarczyk, et al. [25] and Sailer et al. [24] reported that the application of Gluma Desensitizer had a positive effect on the bond strength of SARCs to dentin. On the other hand, Külünk et al. reported that desensitizing agents containing sodium and calcium fluoride reduced the bond strength of an adhesive resin cement to dentin [18]. The results from the present study showed that both immediate and long-term SBS of a SARC to dentin were unaffected by the prior application of the desensitizers. These results can be explained in part by the mechanism of action of the desensitizers used in this study. Desensitizers act through dentinal tubules occlusion. While Gluma^®^ depends on the protein precipitation on the dentinal tubules [29,30,31], Soothe^®^ SDI and Nano P^®^ desensitizers act by forming fluoride compounds precipitates, forming an impermeable film that prevents access to external stimuli, sealing the exposed dentinal tubules [32]. In fact, these mechanisms are responsible for the differences found between Gluma^®^ and Soothe^®^ SDI, as it has been proved that fluor could decrease the bond strength of self-adhesive materials [33]. The present study reveals that, irrespectively of the mechanism of action, the desensitizers could not compromise the bond strength of a SARC to dentin [25]. It has been previously demonstrated that, apart from the fact that SARCs demonstrate limited decalcification/diffusion into dentin [12], the cements thixotropic properties (when applied under pressure) are important to enable SARCs to spread across the entire adhered surface and to establish adhesion [34].

In this study, bond strength values were also tested after storing the specimens for six months to verify the long-term performance of tested materials. In this study, irrespectively of the desensitizer used, SBS of the self-adhesive cement to dentin was statistically lower after six months of aging. This reduction can be explained by the higher hydrophilicity and hygroscopic expansion stress that these materials present [14]. The higher hydrophilicity observed for these materials could be due to the presence of acidic monomers within its composition [35]. Moreover, water sorption of the hydrophilic components of these cements could also accelerate the degradation of ester bonds of some of the resin monomer components, which may have an effect on their mechanical properties, dimensional stability, and biocompatibility, thereby inducing a reduction in the SBS [35]. Moreover, hydrolysis of the adhesive resin layer of self-etch materials has been observed in vitro after water storage, the resulted degradation may be caused by areas of imperfect bonding that are more prone to fluid ingress, and this should be considered during restorative material development. [36]. The composition, the degree of double bond conversion, and the length of the polymer network of these SARCs are related to their physical and mechanical properties. Hence, the effects of the presence of water may affect these properties [37]. The study comparing self-cured and dual-mode SARCs showed equal to slightly inferior flexural and compressive strength of abovementioned materials to other resin cements. The dual-cure cements were slightly stronger than the self-cure cements at 24 h, but when retested after 150-day water storage, there was no difference between the two. This is possibly because the dual-curing cement showed a significant reduction in both flexural and compressive strength after aging [38]. In another investigation where the long-term bond between dual-curing cementing agents was evaluated, it was concluded that dual-curing cementing agents achieve higher bond strength values when light activation is used during polymerization than without light activation [39]. In addition, it could be speculated that, due to the acidic nature of these materials, a limited availability of free radicals and poor polymerization of the material could affect its ability to resist the hydrolytic degradation [40].

It should be highlighted that failure mode analysis revealed that 100% of the failures observed were adhesives. Previous study analyzed the fracture surfaces between SARC and dentin, demonstrating that most of the fails were of the adhesive type [41]. It has been previously stated that the failure mode determines the performance of the applied adhesive systems [42], and in the case of self-adhesive systems, the most likely reason for this behavior is that self-adhesive cements are not aggressive enough to etch beyond the smear layer, thereby severely limiting its mechanical interaction with the dentin [34]. The interface between two dissimilar materials, namely the SARC and the dentin, may be affected by several factors, including contaminants from the operatory field, equipment, or adsorbed environmental contaminants, possibly resulting in a failure location. Under these conditions, water may penetrate the restoration either between and/or through materials, reducing the interaction between the cement and the dentin. Moreover, residual stresses from polymerization kinetics may also disrupt an adhesive bond or limit its durability [43]. This may be the reason why SARCs fractured mainly between resin and enamel or dentin surface and the SBS of SARCs was inferior compared to conventional composite resin cements [44]. 

Changes in the wetting of the dentin surface by the prior application of desensitizers were evaluated by contact angle measurement [45]. Results of the statistical analysis revealed that the water contact angle of dentin was not modified after the application of the different desensitizers. It has been stated that the bond strength between the dentin and the SARC depends on the characteristics and wettability of the adhered surface [46]. Moreover, sealing and bonding characteristics of SARCs mainly depend on their high wettability, which resulted in low microleakage scores at both the cavosurface enamel and dentin surface margins when used as a liner in class II composite restorations [47]. Considering this, the absence of differences in the SBS values observed in this study can be fully explained by this characteristic.

Limitations of the present study include the in vitro design and the fact that only three desensitizers were tested. Further studies are needed to evaluate if these SARCs can adhere to a variety of different substrates, in addition to dentin and enamel, e.g., porcelain and other ceramics, gold, and other metal alloys, as well as indirect composite resins. Moreover, Hardan et al., proved that the use of dentin desensitizer impaired both immediate and aged bond strength [48]. Thus, the effect of the application of these desensitizer agents on demineralized dentin should be evaluated. Additionally, it would be interesting to assess the surface characteristics and the chemical composition after application of different desensitizers (i.e., scanning electron microscopy-energy dispersive X-ray spectroscopy). Moreover, other desensitizing agents and resin cements should be evaluated. Additionally, clinical studies are required to evaluate the performance of these materials in the short and long-term and thus also to evaluate the desensitizing effect and its interaction with SARCs, and in this way to evaluate its limitations and its clinical implications.

## 5. Conclusions

Within the limitations of the present study, it can be concluded that the application of the different types of desensitizers had no significant influence on the SBS of SARCs to dentin. Therefore, this study suggests that the use of these desensitizers before the cementation of indirect restorations with a self-adhesive cement is reliable.

## Figures and Tables

**Figure 1 bioengineering-09-00372-f001:**
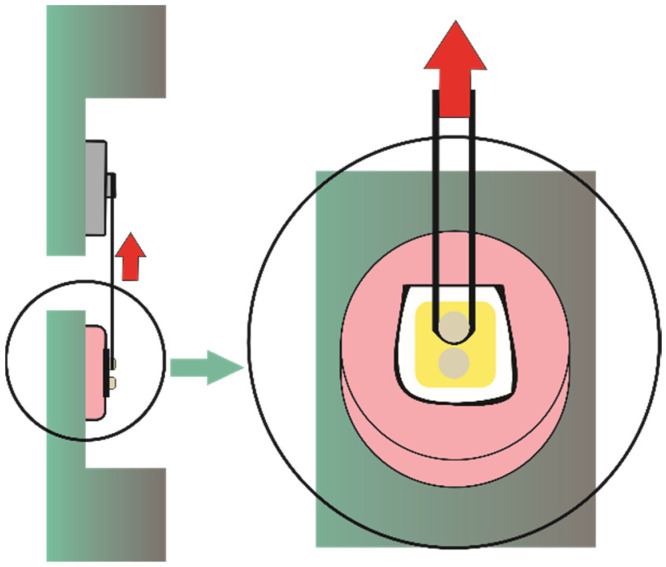
Diagram of the shear bond strength test. Red arrows indicate the direction of the force.

**Figure 2 bioengineering-09-00372-f002:**
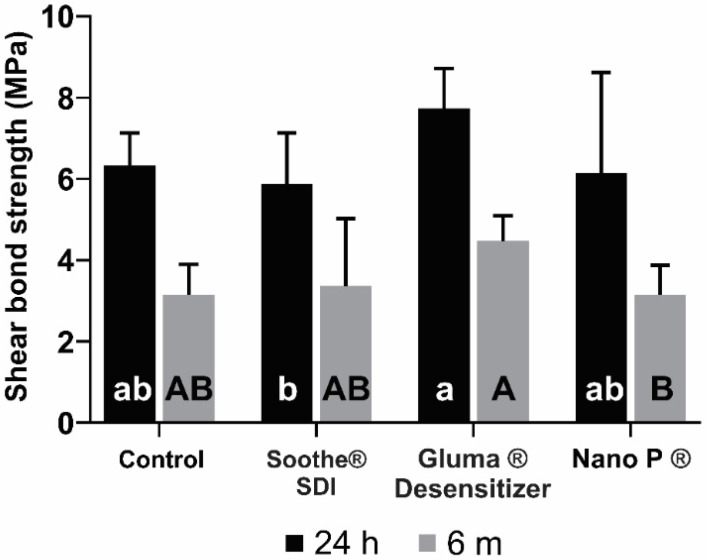
Shear bond strength for study groups (surface treatment and aging times). The same lowercase letters indicate no difference between desensitizers at 24 h. The same uppercase letters indicate no difference between desensitizers at 6 months.

**Figure 3 bioengineering-09-00372-f003:**
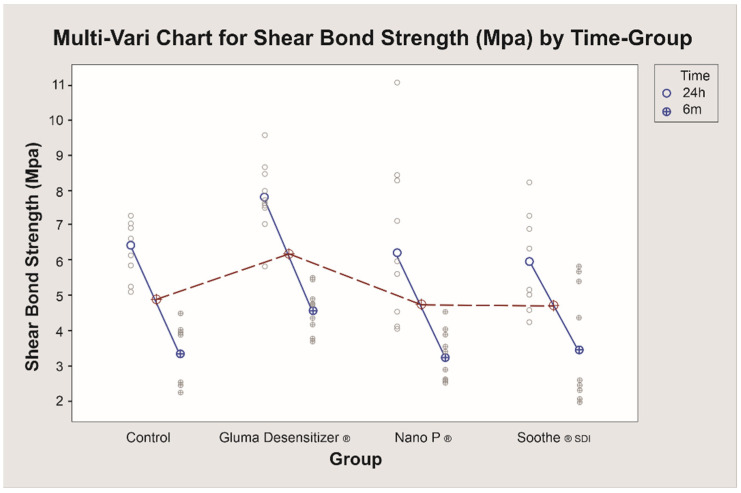
Shear bond strength (MPa) values at 24 h and 6 months. Red circles indicate mean for both aging times.

**Figure 4 bioengineering-09-00372-f004:**
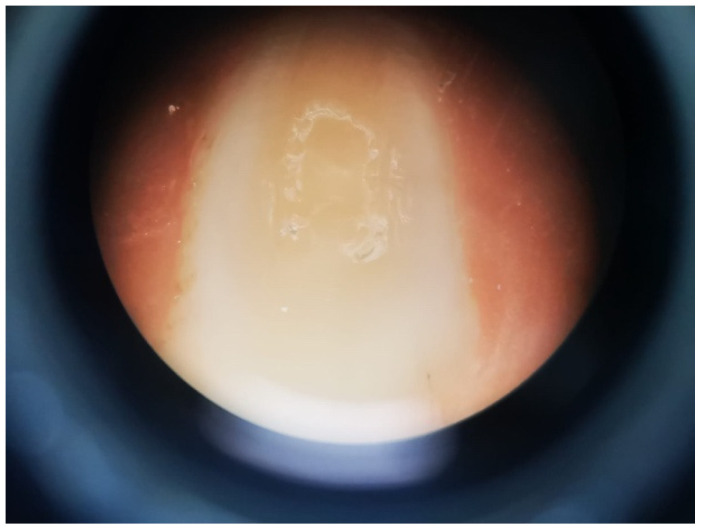
Representative image of the adhesive failure mode of the specimens.

**Figure 5 bioengineering-09-00372-f005:**
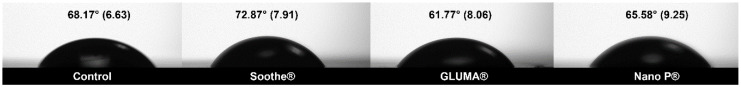
Water contact angle [Mean ± (SD)] for dentin surfaces treated with the desensitizer agents.

**Table 1 bioengineering-09-00372-t001:** Chemical composition and application protocols of the materials used in this study.

Study Group	Desensitizer Agent	Chemical Composition *	Application Protocol
Gluma	Gluma^®^ Desensitizer (Kulzer, Hanau, Germany)	(2-hydroxyethyl) methacrylate glutardialdehyde, purified water	Apply to the dentin for 60 s with a microbrush and dry with dry air until it disappears (observe a non-shiny surface) and then wash with water
Nano P	Nano P^®^ (FGM, Joinville, Brazil)	Potassium nitrate and sodium fluoride	Apply with microbrush on the dentin surface, rub the product with a rubber cup for 10 s, leave the product to rest for 5 min and finally remove the excess with a cotton pad
Soothe	Soothe^®^ (SDI, Victoria, Australia)	6% potassium nitrate and 0.1% fluoride gel	Apply on the surface for 2 min
Cement used: Rely X U200		Base paste: Methacrylate monomers containing phosphoric acid groups, methacrylate monomers, silanated fillers, initiator components, stabilizers, rheological additives Catalyst: Methacrylate monomers, Alkaline (basic) fillers, Silanated fillers, Initiator components, Stabilizers, Pigments, Rheological additives	Mix base paste and catalyst paste into a homogenous paste within 20 s. Spread cement within the restoration and apply moderate pressure

* Information according to the manufacturer’s datasheet.

## Data Availability

The data presented in this study are available on request from the corresponding author.

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
