# Peer review of "Effect of Different Desensitizers on Shear Bond Strength of Self-Adhesive Resin Cements to Dentin"

_bioengineering, 2022, doi:10.3390/bioengineering9080372_

Round 1

Reviewer 1 Report

This study evaluated the effect of different desensitizers on shear bond strength of a self-adhesive resin cement to dentin.

It is original and relevant. It is well written and structured, paragraphs come from each other.  Nevertheless, the are some improvements that need to be addressed before it can be published. 

It should be an explanation of why those aging times (24 h and 6 months) were selected.

In table 1 the column heading “Adhesive protocols” seems to be incorrect, it is suggested to be “application protocols”. Column “Cement” seems to be unnecessary. The protocol for Soothe may be incomplete, since do not refer any wash or removal of the material.

More Information regarding the mechanical test must be provided, since the testing mode of the universal testing machine, was not specified (compressive, tensile, other). The method used to stabilize the samples during the test must be described.

Line 205, results of failure mode are not in accordance with what is proposed in materials and methods section line 174 (type of failure).

In figure 3 ¿what does the number in parenthesis mean? If is the standard deviation it should be indicated and the sing “±” should be added.

This reviewer strongly suggests authors to include images of:

Specimens before testing or mounted for test

A diagram of the applied force direction

A microscopy observation of failure surfaces

Author Response

Dear Sir or Madam,

Thank you for your review. We are very thankful for all your comments that enhanced the quality of manuscript. All your suggestions were addressed accordingly an marked in text.

This study evaluated the effect of different desensitizers on shear bond strength of a self-adhesive resin cement to dentin. It is original and relevant. It is well written and structured, paragraphs come from each other.  Nevertheless, the are some improvements that need to be addressed before it can be published. 

1.It should be an explanation of why those aging times (24 h and 6 months) were selected.

R. Thank you for the comment. The aging times used here were based on the recommendations of the ISO/TS 11405 (Dental materials — Testing of adhesion to tooth structure) international standard. This information was added into the revised version of the manuscript (lines 137-138).

2.In table 1 the column heading “Adhesive protocols” seems to be incorrect, it is suggested to be “application protocols”. Column “Cement” seems to be unnecessary. The protocol for Soothe may be incomplete, since do not refer any wash or removal of the material.

R.We agree with your comment. The heading was changed and the column “Cement” was deleted. Thank you!

3.More Information regarding the mechanical test must be provided, since the testing mode of the universal testing machine, was not specified (compressive, tensile, other). The method used to stabilize the samples during the test must be described.

R.Thank you for the observation. Information was added for clarify this (lines 168-171).

4.Line 205, results of failure mode are not in accordance with what is proposed in materials and methods section line 174 (type of failure).

R.Thank you for the observation. The term “adhesive” was added.

5.In figure 3 ¿what does the number in parenthesis mean? If is the standard deviation it should be indicated and the sing “±” should be added.

R.Yes, the number between parenthesis is the standard deviation. This was specified in the Figure legend. Thank you!

6.This reviewer strongly suggests authors to include images of: Specimens before testing or mounted for test, A diagram of the applied force direction, A microscopy observation of failure surfaces-

R.A diagram of the applied force direction and the microscopy observation of failure surfaces were added into the manuscript. Thank you for the suggestion! (figures 1 and 4).

Reviewer 2 Report

n this manuscript, the authors developed The Study of Effect of Different Desensitizers on Shear Bond Strength of 2 Self-Adhesive Resin Cement is very important to Dentin.

 Their unique inherent properties make them promising candidates for nanomedicine.  It should be published in this journal by addressing the following major queries.

The authors are advised to take into consideration the following suggestions:

1   .  This is a poorly organized manuscript. There are some grammar errors. Please check the main text to improve the English level.

2. Please mention bioactivity material used in this paper such as Resin Cements 

3. The molecular formula of Resin Cements should be added. 

4. Please added morphology results. Please clarify this issue. just SEM or FESEM?

5: Please indicate sizing in this paper .it’s very necessary for Dentin application.

5. Please explain that what is the role of Nano P acts in the synthesis?

6. please add FTIR and XRD to this paper. Therefore, it should be improved for better characterization.

7. Please use updated and recent papers in the literature review to give more sense to the reader.

1: Modification of the epoxy resin mechanical and thermal properties with silicon acrylate and montmorillonite nanoparticles

2: Electrified single‐walled carbon nanotube/epoxy nanocomposite via vacuum shock technique: Effect of alignment on electrical conductivity and electromagnetic interference shielding

3: Anti-bacterial/fungal and anti-cancer performance of green synthesized Ag nanoparticles using summer savory extract

4: Biodegradation study of nanocomposites of phenol novolac epoxy/unsaturated polyester resin/egg shell nanoparticles using natural polymers

5: Bioactive agent-loaded electrospun nanofiber membranes for accelerating healing process: A review

Author Response

Dear Sir or Madam,

Thank you for your review. We are very thankful for all your comments that enhanced the quality of manuscript. All your suggestions were addressed accordingly an marked in text.

In this manuscript, the authors developed The Study of Effect of Different Desensitizers on Shear Bond Strength of 2 Self-Adhesive Resin Cement is very important to Dentin.  Their unique inherent properties make them promising candidates for nanomedicine.  It should be published in this journal by addressing the following major queries.

 The authors are advised to take into consideration the following suggestions:

1.This is a poorly organized manuscript. There are some grammar errors. Please check the main text to improve the English level.

R.Thank you for the comment. The manuscript was checked for grammar errors and reorganized.

2.Please mention bioactivity material used in this paper such as Resin Cements.

R.Thank you for the comment. To the best of our knowledge, resin cements do not possess any bioactivity. If you could guide us regarding this issue we shall be very grateful.

3.The molecular formula of Resin Cements should be added. 

R.Thank you for the suggestion. Information of the composition of the materials used in this study was added into the manuscript (table 1).

4.Please added morphology results. Please clarify this issue. just SEM or FESEM?

R.We added a photo of the dentin surface showing the type of failure. As all the specimens presented adhesive failure type, SEM micrographs are not necessary. Thank you!

5.Please indicate sizing in this paper .it’s very necessary for Dentin application.

R.The specimens’ dimensions were 1.5 mm diameter and 0.5 mm thickness, which resulted in a bonded area of 1.77 mm2. This information was added into the manuscript (line 174). Thank you!

6.Please explain that what is the role of Nano P acts in the synthesis?

R.Nano P acts as a desensitizer through the deposition of nanohydroxyapatite in the dentinal tubule, forming an impermeable film that prevents access to external stimuli. This information was added into the manuscript (lines 259-260). Thank you!

7.please add FTIR and XRD to this paper. Therefore, it should be improved for better characterization.

R.Unfortunately, we are not able to perform this characterization. However, when we get access to these methods we will perform further study on that topic according to you recommendation. We do appreciate your suggestion.

8.Please use updated and recent papers in the literature review to give more sense to the reader.

1: Modification of the epoxy resin mechanical and thermal properties with silicon acrylate and montmorillonite nanoparticles

2: Electrified single‐walled carbon nanotube/epoxy nanocomposite via vacuum shock technique: Effect of alignment on electrical conductivity and electromagnetic interference shielding

3: Anti-bacterial/fungal and anti-cancer performance of green synthesized Ag nanoparticles using summer savory extract

4: Biodegradation study of nanocomposites of phenol novolac epoxy/unsaturated polyester resin/egg shell nanoparticles using natural polymers

5: Bioactive agent-loaded electrospun nanofiber membranes for accelerating healing process: A review

R.Thank you for the comment. We updated the references of the manuscript. Unfortunately, we were not able to add the suggested references since we believe that such references do not fit with the context of the present manuscript.

Round 2

Reviewer 1 Report

None

This manuscript is a resubmission of an earlier submission. The following is a list of the peer review reports and author responses from that submission.